# Cilastatin Modulates DPEP1- and IQGAP1-Associated Neuro-Glio-Vascular Inflammation in Oxaliplatin-Induced Peripheral Neurotoxicity

**DOI:** 10.3390/cells14161294

**Published:** 2025-08-20

**Authors:** Rita Martín-Ramírez, María Ángeles González-Nicolás, Karen Álvarez-Tosco, Félix Machín, Julio Ávila, Manuel Morales, Alberto Lázaro, Pablo Martín-Vasallo

**Affiliations:** 1Laboratorio de Biología del Desarrollo, UD de Bioquímica y Biología Molecular, Universidad de La Laguna, 38206 San Cristóbal de La Laguna, Spain; rmartira@ull.edu.es (R.M.-R.); karenalvtos@gmail.com (K.Á.-T.); javila@ull.edu.es (J.Á.); 2Instituto de Tecnologías Biomédicas, Universidad de La Laguna, 38071 San Cristóbal de La Laguna, Spain; fmachin@fciisc.es (F.M.); mmoraleg@ull.edu.es (M.M.); 3Laboratorio de Fisiopatología Renal, Departamento de Nefrología, Instituto de Investigación Sanitaria Gregorio Marañón, Hospital General Universitario Gregorio Marañón, 28007 Madrid, Spain; mangeleg@ucm.es; 4Departamento de Fisiología, Facultad de Medicina, Universidad Complutense de Madrid, 28040 Madrid, Spain; 5Unidad de Investigación, Hospital Universitario Nuestra Señora de la Candelaria, Instituto de Investigación Sanitaria de Canarias (IISC), 38010 Santa Cruz de Tenerife, Spain; 6Facultad de Ciencias de la Salud, Universidad Fernando Pessoa Canarias, 35450 Las Palmas de Gran Canaria, Spain; 7Departamento de Medicina, Facultad de Ciencias de la Salud, Universidad de La Laguna, 38200 San Cristóbal de La Laguna, Spain; 8Servicio de Oncología Médica, Hospiten Rambla, Grupo Hospiten, 38001 Santa Cruz de Tenerife, Spain

**Keywords:** IQGAP1, dorsal root ganglion (DRG), oxaliplatin, neurotoxicity, allodynia, dehydropeptidase-1, DRG-inflammation, cilastatin, peripheral neuropathy

## Abstract

Oxaliplatin-induced peripheral neurotoxicity (OIPN) represents a major challenge in cancer therapy, characterized by dorsal root ganglia (DRG) inflammation and disruption of neuro-glio-vascular unit function. In this study, we investigated the involvement of the scaffold protein IQ Motif Containing GTPase Activating Protein 1 (IQGAP1) and dehydropeptidase-1 (DPEP1) in the DRG response to oxaliplatin (OxPt) and the modulatory effect of cilastatin. Behavioral assessment showed a robust nocifensive response to cold stimuli in OxPt-treated rats, attenuated by cilastatin co-treatment. Our confocal study revealed different cellular and subcellular expression patterns of IQGAP1 and DPEP1 in neurons, glia, and endothelial cells, where both signals overlap approximately one-third. OxPt enhanced cytosolic aggregation of IQGAP1 in neurons and upregulation of signal in glia, accompanied by co-expression of TNFα and IL-6, indicating involvement in the inflammatory process. DPEP1 showed altered subcellular distribution in OxPt-treated animals, suggesting a potential role in the inflammatory cascade. Notably, IQGAP1 expression was diminished in endothelial membranes under OxPt, while cilastatin preserved endothelial IQGAP1-CD31 colocalization, suggesting partial restoration of blood-nerve barrier integrity. These findings identify IQGAP1 and DPEP1 as key players in DRG inflammation and position cilastatin as a promising modulator of OIPN through neuro-glio-vascular stabilization.

## 1. Introduction

Oxaliplatin (OxPt)-induced peripheral neuropathy (OIPN) is a common and dose-limiting adverse effect of OxPt-based chemotherapy and part of the broader condition known as chemotherapy-induced peripheral neuropathy (CIPN), a form of neuropathic pain resulting from damage to the somatosensory system [1]. Unlike motor neuropathies, CIPN primarily affects sensory neurons, leading to symptoms such as paresthesia, dysesthesia, and tactile deficits [2,3,4,5] and scarcely affects motor neurons [6,7]. OIPN manifests in both acute and chronic forms. The acute phase occurs shortly after OxPt infusion and is transient, often triggered by cold exposure and characterized by jaw tightness, pharyngolaryngeal paresthesia, and cold hypersensitivity. In contrast, the chronic form develops cumulatively and may persist or worsen over time, significantly impairing quality of life and frequently necessitating dose reduction or discontinuation of treatment [8,9,10]. A key pathological feature of OIPN is the preferential accumulation of OxPt in dorsal root ganglia (DRG), facilitated by the absence of a blood-nerve barrier and the presence of fenestrated capillaries. Uptake is mediated via passive diffusion and metal transporters [11]. Then, inside DRG neurons, OxPt disrupts mitochondrial function by altering ion transport, inducing calcium overload, generating reactive oxygen species (ROS), and impairing ATP production, all of which contribute to neuronal apoptosis and axonal degeneration [12,13,14,15]. Despite the clinical impact of OIPN, current treatments for preventing or reversing it remain limited. Understanding the inflammatory mechanisms underlying OIPN is essential to developing effective therapeutic strategies that preserve anticancer efficacy without compromising neurological function.

Recently, we reported the expression and subcellular localization of Dehydropeptidase-1 (DPEP1) in DRG of rats, both in control conditions and six days post-OxPt treatment [16]. The expression of DPEP1 in neurons, glia, and endothelial cells of DRG suggested this protein as a novel target in the prevention of OIPN [16]. Furthermore, in the kidney, DPEP1 inhibition by cilastatin has shown therapeutic potential in renal injury models [17,18]. We demonstrated the usefulness of cilastatin as a nephroprotective agent in an animal model of cisplatin-induced nephrotoxicity, which points to an important relevance of this drug for the preservation of renal function in patients with cancer [17,19].

DPEP1 is a zinc-dependent glycosylated homodimer, originally identified in the brush border of renal proximal tubules, with dipeptidase activity [18,20]. DPEP1 belongs to a family of glycosylphosphatidylinositol (GPI)-anchored membrane-bound enzymes alongside DPEP2 and DPEP3, differing in substrate specificity and tissue distribution [21]. It hydrolyzes diverse dipeptides, including antibiotics like carbapenems, and is involved in glutathione (GSH), leukotriene, and lipid metabolism [20,21,22]. Structurally, DPEP1 features N-glycosylation, disulfide bridges, and Zn^2+^ coordination critical for enzymatic function [23]. In GSH catabolism, DPEP1 degrades Cys-Gly, regulating antioxidant defenses and ferroptosis susceptibility [24,25,26]. Its inhibition by cilastatin protects against nephrotoxicity by preserving GSH and preventing cisplatin-induced ferroptosis [19,27]. DPEP1 also metabolizes leukotriene D4, modulating vascular permeability in the lung, liver, and kidneys [28,29]. Beyond enzymatic roles, DPEP1 acts as a neutrophil adhesion receptor, contributing to inflammatory pathology [29]. DPEP1 overexpression promotes tumorigenesis in colorectal and hepatoblastoma cells via c-Myc and PI3K/Akt/mTOR pathways [30,31]. However, its expression may also suppress invasion in pancreatic and breast cancers [32,33]. Expression varies across tissues—including kidney, lung, liver, and testis—and localizes to membranes, cytosol, and nucleus [21,34]. The role of DPEP1 in the nervous system remains largely unexplored.

A differential screening in patients treated with OxPt between those who suffer from OIPN and those who did not develop it showed IQGAP1 (IQ motif-containing GTPase-activating protein 1) as one of the proteins with higher variation in its expression level [35]. IQGAP1 is a multifunctional scaffold protein that regulates diverse signaling pathways critical for cellular architecture and dynamics [36]. As the best-characterized member of the IQGAP family, IQGAP1 integrates signaling through interactions with over 90 protein partners via its six modular domains, including the calponin homology domain (CHD), IQ domains, and the RasGAP-related domain [36,37,38,39,40]. These structural motifs enable binding to cytoskeletal regulators (e.g., F-actin, N-WASP), kinases (e.g., ERK1/2, B-Raf), small GTPases (e.g., Cdc42, Rac1), and adhesion molecules (e.g., E-cadherin, β-catenin), positioning IQGAP1 as a key mediator of actin dynamics, cell polarity, and MAPK signaling [37,40,41].

Subcellularly, IQGAP1 is primarily localized to the plasma membrane but also functions in the nucleus, where it modulates cell cycle progression and DNA replication [26]. In the nervous system, IQGAP1 is essential for neurite outgrowth [39,42], dendritic spine development, and neurogenesis. Its interaction with Lis1 [43] Rho-family GTPases support neural progenitor migration and differentiation, while decreased IQGAP1 expression impairs neuronal plasticity and may contribute to neurodegenerative and psychiatric disorders [39]. Collectively, IQGAP1 plays a pivotal role in orchestrating cellular processes across various tissues, with particular relevance to neural development and function [40,44,45,46,47]. The passage of neural stem cells and progenitor cells to mature neurons is promoted by vascular endothelial growth factor (VEGF). Balenci et al. [48] suggest that VEGF released by astrocytes is essential in neuronal progenitors’ recruitment to perivascular niches where neuronal differentiation takes place. This function is also related to cell motility activation by IQGAP1 complexes with Rho family GTPases and Lis1 [48]. Finally, IQGAP functions to control dendrite formation, which is related to neuronal plasticity and synaptic input processes [39,49]. Lower levels of IQGAP1 in neurons lead to a decreased number of dendritic tips [50].

Continuing our line of research, we decided to investigate the expression of DPEP1 and IQGAP1 in DRG during OIPN, trying to elucidate their mechanistic interplay and assess the neuroprotective potential of their combined pharmacological modulation by cilastatin in OxPt-treated rats. This study reports behavioral and confocal analyses showing that IQGAP1 and DPEP1 are differentially expressed in neurons, glia, and endothelial cells during OxPt-induced inflammation. OxPts altered their subcellular distribution and promoted co-expression with TNFα and IL-6, implicating them in neuroinflammatory processes. Cilastatin attenuated these effects, suggesting stabilization of the DRG structure-function. These findings present IQGAP1 and DPEP1 as modulators of OIPN and position cilastatin as a potential neuro-glio-vascular protective agent.

## 2. Materials and Methods

### 2.1. Subjects

Animal handling was carried out according to the current legal regulations on the protection of animals used for experimental and other scientific purposes: RD 118/2021, of 23 February; Law 32/2007, of 7 November; and ECC/566/2015, of 20 March. Adult male Wistar rats weighing approximately 250 g were used throughout the OxPt-induced neuropathic allodynia model. These animals were supplied by the Instituto de Investigación Sanitaria Gregorio Marañón animal facility, Hospital General Universitario Gregorio Marañón (HGUGM), Madrid, Spain.

Rats were stabled in conventional cages in pairs, without food/water restriction, stable temperature, and humidity conditions (T = 22 ± 2 °C and HR = 45–65%). Based on recent results showing sex dimorphism in rodents in inflammatory pain regulation and in immune cell signaling in neuropathic pain [51,52], we decided to use males, as other models for chemotherapy-induced peripheral neuropathy (PN) did [53,54,55].

### 2.2. Animal Models

#### 2.2.1. OxPt-Induced Neuropathic Allodynia Animal Model

Animals were classified in four groups: (1) OxPt, *n* = 6, (2) OxPt + cilastatin, *n* = 6, (3) control, *n* = 4, and (4) cilastatin, *n* = 4. OxPt was supplied by the HGUGM Pharmacy Service at an initial concentration of 5 mg/mL dissolved in 5% glucose solution (Braun Medical S.A., Barcelona, Spain) and administered at a final concentration of 6 mg/kg, in a single dose intraperitoneal linjection, at the beginning of the study that lasted 6 days. The control group received an injection with the same vehicle, in the same conditions and volumes as the treated groups. Cilastatin, an inhibitor of the renal enzyme DPEP1, was supplied by ACS DOBFAR (s.p.a., Tribiano, Milan, Italy). The concentration used was 150 mg/kg dissolved in a volume of 0.5 mL of 0.9% saline solution. Cilastatin was injected intraperitoneally, immediately after OxPt administration, and every 24 h. The dose and administration regimen of cilastatin were established by literature review and by the group’s experience [56]. The rats were weighed daily to adjust the dose of cilastatin and to check their behavior.

On day 6, euthanasia took place. The rats were anesthetized with sevoflurane (Abbvie, Madrid, Spain) at 5% and maintained during surgery at 2%. Once the rats were weighed and placed in the surgical field, they were opened longitudinally, removing only the first layer of skin at the level of the sternum for blood extraction by cardiac puncture. Using scissors, the hair was parted along the spine in a distal direction, and the spine was isolated by cutting on both sides, along the spine beyond the pelvic bone. With the aid of a magnifying lens and microsurgical material, the dorsal (lumbar) root ganglia and the sciatic nerve were located and removed. These ganglia were placed on a dark surface for better manipulation, and a drop of paraformaldehyde (PFA) was added. They were then placed in cassettes with PFA for 24 h. Finally, samples were placed in 70% ethanol (VWR, Radnor, PA, USA). After this, other tissues and organs such as the liver, colon, and kidney were collected and stored at −80 °C for further analysis.

No mortality, diarrhea, or signs of alopecia were observed in any group of animals during the study. Rats were weighed before chemotherapy administration to record baseline weight and on each day of the study. No significant increase or decrease in body weight was observed in the rats during the study.

#### 2.2.2. Allodynia Test

The acetone test was used to evaluate cold allodynia by touching the plantar skin of hind paws with a 200 μL droplet of acetone (PanReac, Barcelona, Spain) [57] from an insulin-type syringe (B. Braun Medical S.A., Madrid, Spain). Dripping acetone was performed on only one hind paw at a time, alternating the left and right sides each time with the intention of preventing habituation. Every day, the times of flicking, biting, or licking the stimulated paw were counted for 2 min. The times registered were the median of testing three times at a 1 h interval.

### 2.3. Antibodies

Primary and secondary antibodies used for immunohistochemistry are listed in Table 1.

### 2.4. Western Blot

After homogenizing, tissue samples (kidney, liver, and colon) were boiled in LSB buffer at 95 °C for 10 min. The resulting whole cell lysates were separated on a 4–20% denaturing polyacrylamide gel and transferred onto Immobilon™-158 membranes (Millipore) via electroblotting. Membranes were then blocked for 1 h in PBS containing 5% BSA. Proteins were detected using primary antibodies against DPEP1 or IQGAP1 (refer to Table 1), followed by a horseradish peroxidase-conjugated anti-rabbit IgG secondary antibody (Cat. # NA9340, GE Healthcare, Madrid, Spain). Visualization was carried out using the ChemiDoc XRS imaging system (Bio-Rad Laboratories, Hercules, CA, USA) with the Immobilon Western Chemiluminescent HRP substrate (Merck Millipore, Darmstadt, Germany), according to the manufacturer’s protocol. Laminin and α-Actin were used as loading controls.

### 2.5. Immunohistochemistry

DRG tissue samples were embedded in paraffin and cut into five-micron-thick sections. Tissue sections were deparaffinized in xylene and rehydrated in a 100%, 96%, and 70% alcohol bath, sequentially. Epitope retrieval was performed by heating samples in sodium citrate buffer (pH 6.0) at 120 °C for 10 min in an autoclave. Then, non-specific sites were blocked with 5% BSA or serum in Tris-buffered saline (TBS) for 1 h at room temperature.

Finally, for the immunofluorescence staining, tissue sections were incubated with primary antibodies (see Table 1) overnight at 4 °C. Samples incubated without primary antibodies were used as a negative control. Slides were incubated for 1 h at room temperature in the dark with secondary antibodies raised in different species and conjugated to different fluorochromes. Slides were mounted with ProLong^®^ Diamond Anti-fade Mountant with DAPI to visualize cell nuclei.

### 2.6. Confocal Analysis

Slides were analyzed, and digital images were captured using Zeiss LSM980 Airyscan-2 (Zeiss, Oberkochen, Germany) and Leica SP8 (Leica Microsystems, Wetzlar, Germany) confocal microscopes. Raw images in Carl Zeiss Image Data File (CZI) or Leica Image Format (LIF) were exported as Joint Photographic Experts Group (JPEG) at 300 dots per inch (dpi). Figures were assembled using Adobe Photoshop CC 2018 and exported at 300 ppi.

### 2.7. Image Quantitative Analysis

Image analysis was carried out using ImageJ software, version 1.54p (National Institutes of Health; Bethesda, MD, USA) with the EzColocalization plugin (10.1038/s41598-018-33592-8). Confocal images used for analysis were taken using the same parameters. Laser power and detector gain settings were optimized to cover fluorescence signals in a 16-bit depth range without saturation. Changes in fluorescence from baseline were measured as the mean intensity of selected regions of interest; 50 measurements (at least) were made per photograph in four photographs (at least) of different fields of each different immunostaining. Complementarily, the “Cell Counter” plug-in was used to ensure that neurons or glia were counted only once. The Pearson correlation coefficient (PCC) was used to quantify the degree of colocalization between fluorophores [58,59].

### 2.8. Image Qualitative and Semi-Quantitative Analysis

Although in this report we will only present the results of the quantitative analysis, being aware of the possible errors in the use of plugins [58], we performed a blind qualitative and semi-quantitative analysis, which we then compared with the quantitative analysis specified below in the quantitative analysis section. The analysis was conducted as follows: Two independent observers evaluated preparations and photographs blindly, grading the staining intensities as absent (–), faint (+), moderate (++), or strong (+++). Cutoffs were established by consensus among observers after an initial review of several samples of varying appearance and blind coding. In cases where the score data differed by more than one unit, the means of the score data were calculated. A table with the results of this analysis is included in the Appendix A section.

### 2.9. Statistical Analysis

SPSS version 29.0 for Windows (IBM Corp., Armonk, NY, USA) was used for statistical analysis. Values were subjected to two-way ANOVA using means per day, followed by the Newman–Keuls test; acetone test data for allodynia are presented as mean  ±  standard error of the mean (SEM). Values were subjected to analyses. A probability value (*p*) < 0.05 was considered statistically significant. Quantification of immunofluorescence intensity was performed for each specific immunohistochemical marker corresponding to the analyzed proteins. The non-normal distribution of the data was confirmed using the Kolmogorov–Smirnov test. Differences in staining intensity between the control and specific treatments were assessed using the non-parametric Kruskal–Wallis test [60]. A *p*-value of < 0.05 was considered statistically significant. To determine the significance, quantitative variables were summarized as the mean ± SEM. Equality of variances was tested with Levene’s test. Normally distributed continuous variables with equal variances were analyzed with analysis of variance, and Student’s t-test was used to test the difference between the responses of two groups (IQGAP1-DPEP1, -MAP2, -CD31, -GFAP, TNFα, IL-6, and any other reported in this study) as a means test of the points obtained; if variances were not equal, the Kruskal–Wallis test was performed. *p* < 0.05 was considered statistically significant.

## 3. Results

### 3.1. Differential Nocifensive Response to Allodynia Among Controls, OxPt-, and Cilastatin-Treated Rats

Nocifensive response to drops of cold acetone stimulation was evaluated every 24 h, at the same time during the day, over the six days of the study, by an observer unaware of the experimental conditions of the rat group. The number of times the animal bit, flicked, or licked the examined paw increased by a factor of two to fifteen in rats treated with OxPt compared to those in the control and control + cilastatin groups (Figure 1B). Control and cilastatin-treated rats did not show any considerable movement; this number varied from 8 to 18 in rats undergoing OxPt and only 3–6 times/2 min in the animals pretreated with cilastatin and undergoing OxPt. Significant differences (*p* < 0.05) were found from the second day on between the OxPt- and OxPt + cilastatin-treated groups, showing a protective effect of cilastatin from OxPt-elicited allodynia.

### 3.2. Tissue-Specific Expression Patterns of DPEP1 and IQGAP1 by Western Blot Reveal the Need for Cellular-Level Analysis in Peripheral Neuropathy

Figure 2 shows the expression of DPEP1 and IQGAP1 in two or three different samples of homogenates of colon and kidney from rats of the different studied groups (control, cilastatin, OxPt, and OxPt plus cilastatin). The quantification showed slight variation or no variation in intensity. However, when these tissues were studied by immunohistochemistry, remarkable differences were found among cells and tissue structures (renal glomerulus, tubules) [61]. Due to this fact, we decided to perform a confocal study using specific and well-characterized antibodies in order to find specific effects in subpopulations of cells among those of DRG, the most affected organ in PN.

### 3.3. Immunohistochemical-Confocal Localization of DPEP1 and IQGAP1 in DRG from OxPt- and Cilastatin-Treated Rats

In samples from rats of the control group, immunofluorescence signal for DPEP1 was found in the cytosol of neurons, ranging from medium to high intensity and following a reticulated pattern, and in satellite glial cells at a slightly higher intensity level, Figure 3, upper lane. As for IQGAP1, it exhibited specific immunofluorescence at a high intensity level localized in the plasma membrane of most neurons, and in cytosol at a much lower level in a homogeneously granulated fashion. In satellite glial cells, IQGAP1 immunofluorescence was present in the plasma membrane at high levels.

DPEP1-specific immunofluorescence signals in cilastatin samples were homogeneously distributed at medium intensity in neurons and glial cells, resulting in a reticulated pattern (magenta stars in the second lane of Figure 3). The IQGAP1 immunofluorescence signal was ubiquitously localized and in a granulated pattern (yellow asterisks) as in the control panels. Furthermore, a higher-intensity red signal was present in the plasma membrane of neurons and satellite cells (white arrowheads).

OxPt panels in Figure 3 display representative images of DPEP1-specific immunofluorescence signal within neurons, in a similar reticulated pattern to that of images above (yellow asterisks) and slightly brighter in the membrane of neurons (white arrows). IQGAP1-specific immunostaining signal panels depicted higher intensity compared to that of the control in satellite cells (yellow arrowheads). The signal was mainly present near the plasma membrane and at a lesser intensity in body cells. High-intensity fluorescence specific for IQGAP1 displayed a spot pattern in the cytosol of some neurons (magenta arrowheads).

OxPt + cilastatin panels, Figure 3, showed a reticulated pattern of DPEP1 signal, presented with higher intensity in a few neurons (red arrowheads); white stars in images mark a more homogeneous distribution of DPEP1 immunofluorescence signal within neurons. A high level of IQGAP1 immunostaining signal was evident in the plasma membrane of glia cells and neurons (yellow arrows). There was no specific signal of IQGAP1 inside axons from the root and in certain neurons (yellow stars).

### 3.4. IQGAP1 Expression in Glial Cells in DRGs from OxPt- and Cilastatin-Treated Rats

In order to learn the effects of OxPt and cilastatin on IQGAP1 of DRG glial cells and to discriminate them from other cells similar in localization and size, such as endothelial cells, co-labeling for this scaffoldine and GFAP was performed. Figure 4 shows exemplary confocal microscopy images of immunostaining for IQGAP1 and GFAP. In control samples, the IQGAP1 signal showed the typical pattern in neurons, as described in Figure 3. Merge image in control shows a high IQGAP1-GFAP co-labeling level in presumably activated GFAP-specific fluorescence in samples from cilastatin- or OxPt + cilastatin-treated rats, apparently and quantitatively lower than in samples from animals that did not undergo cilastatin.

### 3.5. IQGAP1 Expression in Endothelial Cells of DRG

Co-immunostaining for IQGAP1 (green) and CD31 (red) in Figure 5, control panels display IQGAP1 immunofluorescence representative images in neurons following the distribution and expression patterns described in previous figures and in this Section 3. The IQGAP1 signal of brighter intensity around neurons corresponded mainly to satellite glial cells (yellow arrows). The CD31 immunostaining signal identified endothelial cells near neurons (magenta arrowheads) and surrounding axons of the root, where a specific signal for IQGAP1 at a medium intensity level was evident in CD31-positive cells.

In cilastatin-treated rats, a higher intensity of the immunostaining signal for IQGAP1 was present in the membranes of glia cells and neurons, and a few of the highest intensities were present in plasma membranes, even in axons of the root. A faint CD31 signal was shown in the ganglion, with the exception of some points. The immunofluorescence signal of CD31 was brighter around axons in the root (yellow stars).

In samples from OxPt-treated rats, neurons presented an IQGAP1-specific bright signal in the cell periphery, touching the plasma membrane, and a lower intensity in the cytosol, exhibiting a reticular pattern. A higher level of IQGAP1 signal was present in some satellite glial cells in a dot pattern (white arrowheads). The CD31 immunostaining signal was found at medium/high intensity in the root, at a lower intensity level. Co-immunolabeling between CD31 and IQGAP1 was not present in endothelial cells at the root of axons in merged images.

When rats were treated with cilastatin and OxPt at the same time, a high-intensity immunofluorescence signal for IQGAP1 was in the cytosol of glia cells, depicting a reticular pattern, and at a lower level was present in the cytosol of neurons in a reticulated fashion. In comparison with the other groups of images, the CD31 immunostaining signal evidenced endothelial cells around axons of the root (magenta arrows), and some endothelial cells in the ganglion body, near neurons, showed medium/low intensity of red immunofluorescence signal. Some evidence of co-labeling between IQGAP1 and CD31 signals was found.

### 3.6. IQGAP1 in the Short-Time DRG Inflammation Response to OxPt Chemotherapy

Short-term involvement of IQGAP1 in the inflammation response was checked by co-labeling with TNFα (Figure 6). In control samples, a faint or no signal for TNFα was found in neurons. IQGAP1-specific fluorescence presented the typical picture of heterogeneous intensity with higher intensity in the plasma membrane of most neurons and, at a much lower level, in the cytosol, in a reticular and grainy background. A similar pattern was observed in satellite glial cells surrounding neurons. In cilastatin, IQGAP1 and TNFα co-staining displayed distribution and intensity patterns of fluorescence signals similar to those of controls. However, in OxPt panels, a higher intensity of immunostaining signal for IQGAP1 in the cytoplasmic membrane and in the cytosol than in control panels was seen, along with a high-intensity immunofluorescence-specific signal for TNFα following a dot pattern inside neurons (magenta arrowheads). Merges portray a heterogeneous fashion of co-expression, mainly in the cytosol. Interestingly, OxPt + cilastatin panels show specific fluorescence signals for either IQGAP1 or TNFα, similar to those of control and control + cilastatin.

Representative images of the immunostaining for IQGAP1 and IL-6 are shown in Figure 7. In control panels, immunofluorescence signal for IQGAP1 was found at heterogeneous intensity levels varying from medium to high, brighter in the cytoplasmic membrane and peripheral areas of cytosol, showing the cell’s shape in neurons, satellite glial, and capillaries, and leaving a separation interface between the cells. A low-intensity signal for IL-6 immunostaining was shown in controls. Medium- to low-intensity signal was found in satellite cells where IQGAP1 and IL-6 signals colocalize in endothelial capillary cells (erythrocytes inside, yellow stars). IQGAP1 and IL-6 specific signals in DRG samples from cilastatin-treated rats were similar in intensity and localization to those shown in control panels. Cells in OxPt panels portray photographs with heterogeneous signals for IQGAP1, ranging within the medium intensity level, not as bright as in control or cilastatin samples. Quantitation demonstrated a higher-intensity signal for IQGAP1 inside glia cells in this group compared to the control. IL-6 and IQGAP1 signals colocalized to a great extent. As in the case of IQGAP1-TNFα OxPt + cilastatin samples, IQGAP1-specific immunofluorescence was present in a heterogeneous intensity pattern, mainly in neurons displaying a high-intensity signal in the cytoplasmic membrane and neighboring areas of cytosol, very similar to that in control panels. The signal for IL-6 was at a moderate intensity level.

Figure 8 displays the compilation of results of the expression and localization of TNF-α, DPEP1, and IQGAP1 in DRG neurons, glia, and endothelial cells in control, cilastatin-, OxPt-, and OxPt–cilastatin-treated groups of rats and statistical significance.

## 4. Discussions

This study identifies IQGAP1 and DPEP1 as participants in DRG inflammation in OIPN; concurrently, OxPt altered their subcellular distribution in neurons, glia, and endothelial cells, promoting inflammatory signaling and neuro-glio-vascular disruption. Notably, cilastatin attenuated nocifensive responses and prevented inflammation and allodynia, suggesting partial morpho-functional integrity. These findings highlight the mechanistic involvement of IQGAP1 and DPEP1 in OIPN and propose cilastatin as a potential therapeutic modulator targeting neuro-glio-vascular dysfunction.

The graph in Figure 1B shows how the administration of cilastatin together with OxPt effectively prevents allodynia in the same way that it protects renal function [17,61] and, at the same time, confirms the involvement of DPEP1 in the neuropathic and inflammatory process.

Differential immunofluorescence patterns of DPEP1, IQGAP1, TNFα, and IL-6 across treatment groups highlight the pleiotropic role of IQGAP1 in the context of OxPt-induced DRG inflammation, as well as the modulatory capacity of cilastatin. IQGAP1 is involved in signal transduction and cytoskeletal and membrane dynamics (specified in the Section 1) performing variable functions depending on cell type (neurons, GFAP+ glia, and CD31+ endothelial cells) and modulating the effects according to the treatment conditions, suggesting a multifaceted role in the neuroinflammatory response and confirming, this way, the qualifying of the puppeteer [21].

IQGAP1 was predominantly located in the plasma membrane and pericytosolic areas in DRG neurons from control and cilastatin-treated animals, displaying a granular and reticulated pattern, in accordance with its function in organizing cortical actin and stabilizing cell polarity. Upon OxPt treatment, IQGAP1 distribution was altered significantly, showing enhanced cytosolic dot-like aggregates and intensified membrane labeling, indicating a stress-induced response of IQGAP1 associated with neuronal cytoskeletal remodeling and impaired trafficking or reactive plasticity. Interestingly, the co-localization with TNFα and IL-6 in the cytosol of neurons supports a role of IQGAP1 in inflammation, probably linked to cytoskeletal perturbations, a glimpse into neuroinflammatory processes. Ghosh et al. [62] showed the link between the rise in pro-inflammatory molecules, such as TNFα, and the rearrangement in cytoskeleton structure under oxidative stress in glioma cells. There were no changes in quantitative expression of IQGAP1, though intensity of interaction with other molecules was described, as, e.g., Cdc42 [62], a protein from the Rho family, activated via IQGAP1, which contributes to its translocation to the plasma membrane, where it regulates actin polymerization [63].

On the other hand, changes in subcellular localization of this scaffold protein have been previously described in tumoral cells; Rotoli et al. [64] showed an altered expression of IQGAP1 in colorectal carcinoma cells in comparison to healthy samples in which increased levels of IQGAP1 were found around plasma membranes, probably related to modifications in cell polarity in line with tumoral progression [65,66].

IQGAP1 has a key role in cytoskeleton structure and rearrangements in cells under stress conditions, intimately related to different signal pathways activation as a result of inflammatory responses [21]. Our results showed that this function of IQGAP1 is also activated in neurons and could contribute to OIPN.

In satellite glial cells (GFAP+), OxPt induced a clear upregulation of IQGAP1, particularly in perinuclear regions and membrane domains, suggesting reactive gliosis and potential involvement of IQGAP1 in glial activation and neuro-glial crosstalk. The strong co-expression with TNFα and IL-6 in this context further supports IQGAP1 participation in amplifying the inflammatory microenvironment through glial cells. The two subpopulations of TNFα, in the cytosol and at the plasma membrane, could indicate an increase in biosynthesis and an increase in extracellular binding to its membrane receptor. Interestingly, cilastatin significantly attenuated these alterations, restoring IQGAP1 patterns in glia to those seen in controls and reducing cytokine co-expression, which may reflect a protective effect of cilastatin by preserving glial homeostasis and limiting neuro-glial pro-inflammatory signaling pathways. Effects of oxidative stress in glial cells have been previously described [67] and were related to GFAP expression and cytoskeleton reorganization leading to cell death, but this is the first time that IQGAP1 expression is studied in these cells.

Endothelial cells (CD31+), although less explored in neuropathic models, also showed intriguing behavior. In OxPt-treated DRGs, while CD31 immunostaining was preserved around axons, IQGAP1 was absent in endothelial membranes, contrasting with its presence in controls, suggesting a vascular disassembly or dysfunction possibly contributing to the neuroinflammatory milieu. Importantly, cilastatin partially restored IQGAP1 expression in endothelial cells, indicating a potential endothelial-protective effect of cilastatin that could contribute to maintaining blood-nerve barrier integrity. Two studies from Yamaoka-Tojo et al. addressed this topic and found a relationship between reduction in cell adhesion through endothelial cells and promotion of angiogenesis mediated by IQGAP1 and vascular endothelial (VE)-cadherin [68,69]. IQGAP1 demonstrated a dual function in endothelial cells; in basal conditions, it has a role in adherence junctions and maintaining cell-to-cell adhesion; however, under stress circumstances and increased levels of ROS, IQGAP1 promotes phosphorylation of VE-cadherin, loss of cell-to-cell contact, and VEGF2 recruitment [68,69,70].

The heterogeneity in the expression of markers, as well as of IQGAP1 and DPEP1 in neurons and glia, leads to the possibility of polymorphisms; however, to our knowledge, no association studies exist between IQGAP1 expression or protein variants and OIPN. However, different basal levels of IQGAP1 (often regulated through polymorphisms either in the gene itself or regulatory microRNAs) have been linked to human cognitive performance and multiple sclerosis [71,72]. As for DPEP1, only a clear link between gene variants and osteoarthritis has been established [73]. In addition, DPEP SNPs have been associated with changes in plasma homocysteine levels in healthy women, which epidemiologically is linked to later development of cardiovascular diseases and loss of cognitive function [74]. Finally, other Genome-Wide Association Study (GWAS) and meta-analysis studies have found associations between DPEP1 variants and other traits (plasma and urine metabolites, blood pressure, etc.) [75]. Thus, it is indeed plausible that IQGAP1 and/or DPEP1 genetic variants regulate their expression, location, and/or responsiveness upon oxaliplatin in DRG.

Collectively, these data support a pleiotropic role of IQGAP1, acting not only as a cytoskeletal organizer but also as an integrator of inflammatory signals within the DRG microenvironment, modulating neuronal, glial, and endothelial compartments in a cell type-specific and stimulus-dependent manner. Cilastatin appears to modulate these responses by preserving physiological patterns of IQGAP1 localization and attenuating the aberrant inflammatory cascade induced by OxPt. This positions IQGAP1 as a critical node in the neuro-glio-vascular unit dysfunction during chemotherapy-induced peripheral neuropathy and cilastatin as a potential modulator of these pleiotropic effects.

## Figures and Tables

**Figure 1 cells-14-01294-f001:**
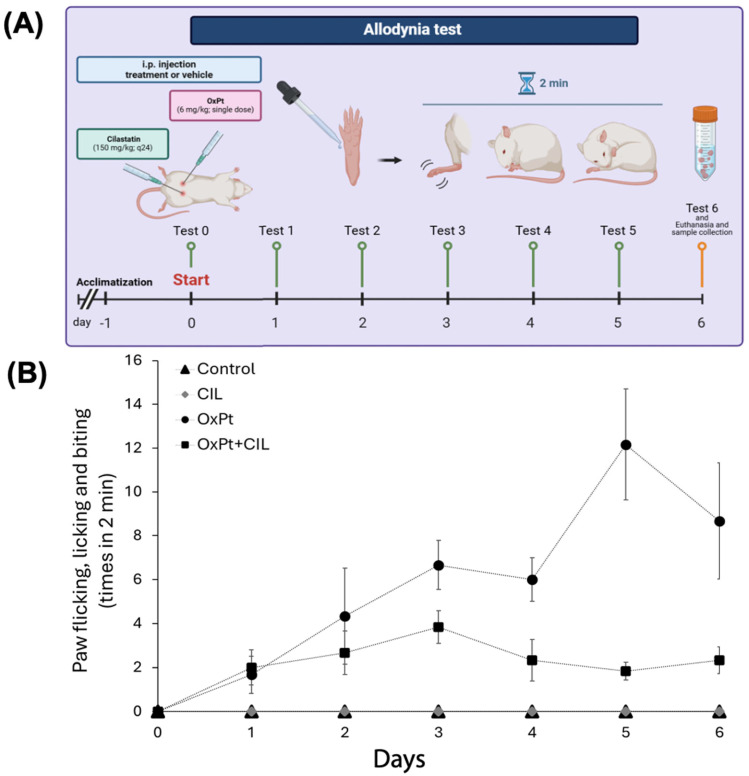
Workflow of the model generation. (**A**) Cold allodynia was tested with the acetone test performed on all six days by touching the plantar skin of both hind paws with a 200 μL droplet of acetone. The times of flicking, licking, or biting the stimulated paw were counted for 2 min. The times were considered as the average of testing three times at a 1 h interval. (**B**) The allodynia test was run on control, OxPt, cilastatin (CIL), and OxPt + CIL rats every 24 h. The data are expressed as the mean ±SEM. Blue asterisks indicate statistical significance of the OxPt group compared to all other groups (*p* < 0.01). The red asterisk indicates statistical significance compared to the Control ± CIL group, but not compared to the OxPt + CIL group. There is a significant difference (*p* < 0.05) between controls and OxPt- and OxPt + cilastatin-treated groups from the second day.

**Figure 2 cells-14-01294-f002:**
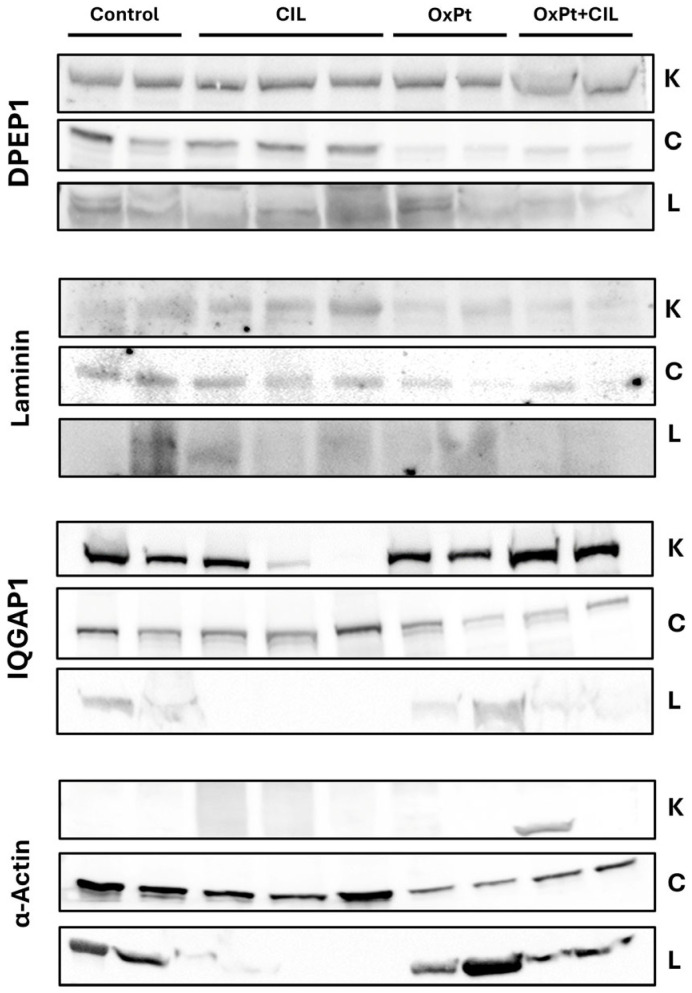
Western blot analysis probed with DPEP1C and IQGAP1 antibodies in rat kidney (K), colon (C), and liver (L) from control, cilastatin (CIL)-, OxPt-, and OxPt + CIL-treated groups. Samples are from two (Control, OxPt, and OxPt + CIL) or three (CIL) different animals. Data were analyzed using a mixed-effects model. For DPEP1 in K, a significant difference was observed between the OxPt + CIL and Control groups, *p* < 0.0378. No statistically significant differences were found in C or L. IQGAP1: No statistically significant differences were detected in K, C, or L.

**Figure 3 cells-14-01294-f003:**
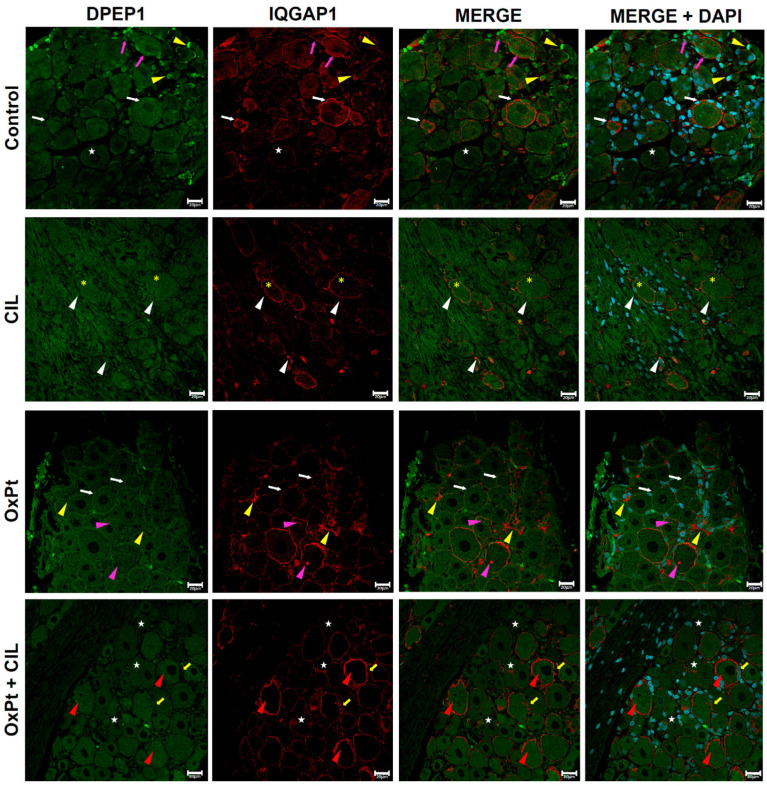
Immunostaining for DPEP1 (green) and IQGAP1 (red). Control DPEP1 neurons cytosol signal (white stars), reticulated pattern, in glia (yellow arrowheads). IQGAP1 neuronal plasma membranes (white arrows), faint and granular in cytosol (yellow asterisks), and glial plasma membranes (magenta arrows). Cilastatin: DPEP1 in neurons and glia (magenta stars). IQGAP1 granular cytosolic signal (yellow asterisks), plasma membrane labeling (white arrowheads). OxPt: DPEP1 reticulated cytosolic pattern (yellow asterisks), brighter in neuronal membranes (white arrows). IQGAP1 increased in satellite glia cells (yellow arrowheads), near the plasma membrane, as cytosolic dots in some neurons (magenta arrowheads). OxPt + CIL: DPEP1 reticulated, increased intensity in some neurons (red arrowheads), and homogeneous distribution (white stars). IQGAP1, a strong membrane signal in neurons and glia (yellow arrows), is absent in axons and some neurons (yellow stars). Scale bar 20 µm. (*n*-Control = 4, 178 photographs, 7 DRGs; *n*-CIL = 4, 184 photographs, 8 DRGs; n-OxPt = 6, 215 photographs, 10 DRGs; *n*-Ox-Pt + CIL = 6, 185 photographs, 7 DRGs). OxPt increased IQGAP1 expression in neurons and glia 2–3-fold, with respect to control. *p* < 0.05, within a heterogeneous population. Magnified views of highlighted regions are available in Appendix A.

**Figure 4 cells-14-01294-f004:**
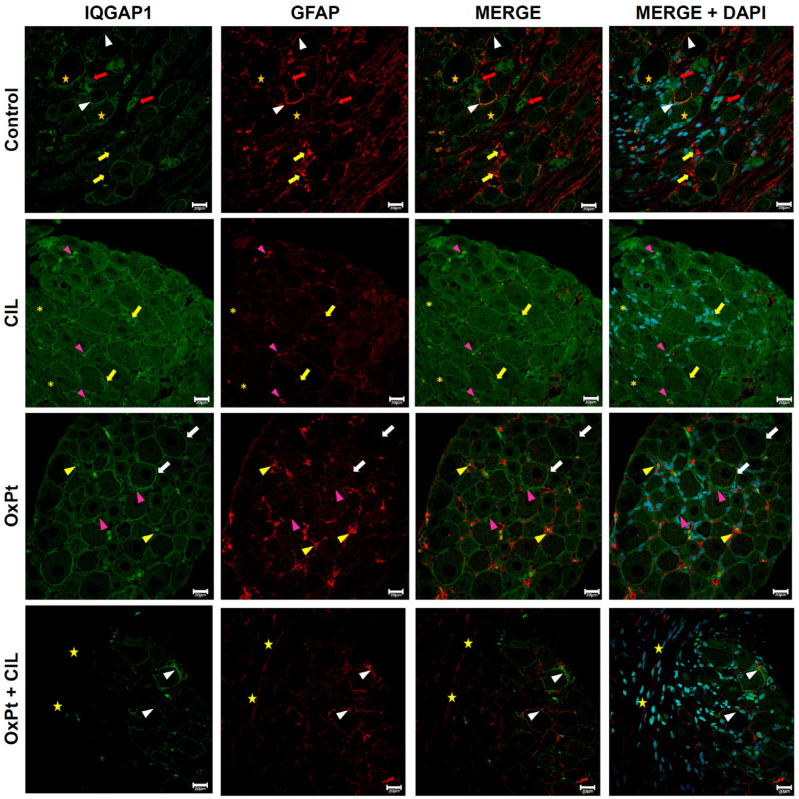
Immunostaining for IQGAP1 (green) and GFAP (red). IQGAP1 is intense in membranes and lower in neuronal cytosol (orange stars). GFAP in glia (white arrowheads), absent in axons (magenta arrowheads), and higher levels in activated glia (yellow arrows). Glial IQGAP1 without GFAP (red arrows). Cilastatin: IQGAP1 increased in neurons (white stars) and colocalized with GFAP in glia (magenta arrowheads, yellow arrows). GFAP at low–medium intensity, dot-like, and present in root glia (yellow asterisks). OxPt: IQGAP1 has strong fluorescence in neuronal membranes (white arrows), lower in cytosol (yellow stars), and increased in glial membranes (magenta arrowheads). GFAP is strong in activated glia lacking IQGAP1 (yellow arrowheads). OxPt + CIL: IQGAP1 is intense in neuron/glia membranes and low in axons (yellow stars), with a medium/low cytosolic pattern (yellow stars). IQGAP1/GFAP colocalization is evident (white arrowheads). Root glial GFAP is low/medium (magenta asterisks). Scale bar 20 µm. (*n*-Control = 3, 163 photographs, 5 DRGs; *n*-CIL = 3, 204 photographs, 7 DRGs; *n*-OxPt = 3, 200 photographs, 8 DRGs; *n*-Ox-Pt + CIL = 2, 178 photographs, 5 DRGs). OxPt increased GFAP expression in glia with respect to CIL and OxPt + CIL, *p* < 0.05. See Appendix A for zoomed-in images of selected areas.

**Figure 5 cells-14-01294-f005:**
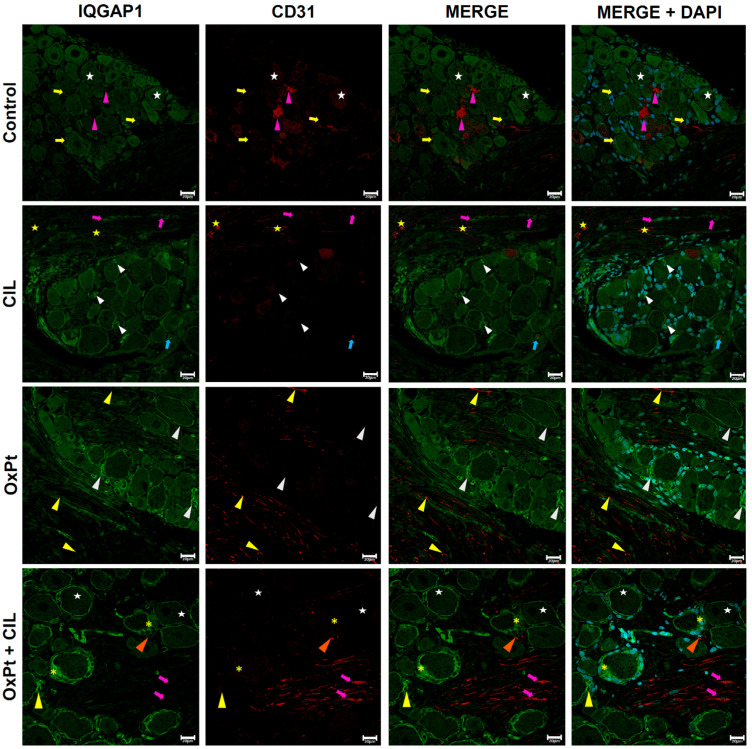
Co-immunostaining of IQGAP1 (green) and CD31 (red) in DRG endothelial cells. Control: IQGAP1 neurons (white stars), glia (yellow arrows), and endothelial cells near neurons (magenta arrowheads) with CD31 signal. Cilastatin (CIL): IQGAP1 is higher in glial membranes (white arrowheads) and root (magenta arrows). CD31 is faint except near axons (blue arrow, yellow stars). OxPt: IQGAP1 is bright at neuronal periphery and dot-like in glia (white arrowheads). CD31 at medium/high intensity in the root, no colocalization with IQGAP1 (yellow arrowheads). OxPt + CIL: IQGAP1 has a reticulated pattern in glia (yellow asterisks) and in neurons (white stars). Endothelial cells CD31+ near axons (magenta arrows) and neurons (orange arrowheads), some co-labeling with IQGAP1 (yellow arrows, yellow asterisk). Scale bar 20 µm. (*n*-Control = 4, 155 photographs, 4 DRGs; *n*-CIL = 4, 190 photographs, 4 DRGs; *n*-OxPt = 6, 175 photographs, 9 DRGs; *n*-Ox-Pt + CIL = 4, 185 photographs, 6 DRGs). OxPt increased IQGAP1 expression in endothelial cells of roots with respect to control and CIL (*p* < 0.05). No significant OxPt vs. OxPt + CIL. Detailed close-ups of specific regions are provided in Appendix A.

**Figure 6 cells-14-01294-f006:**
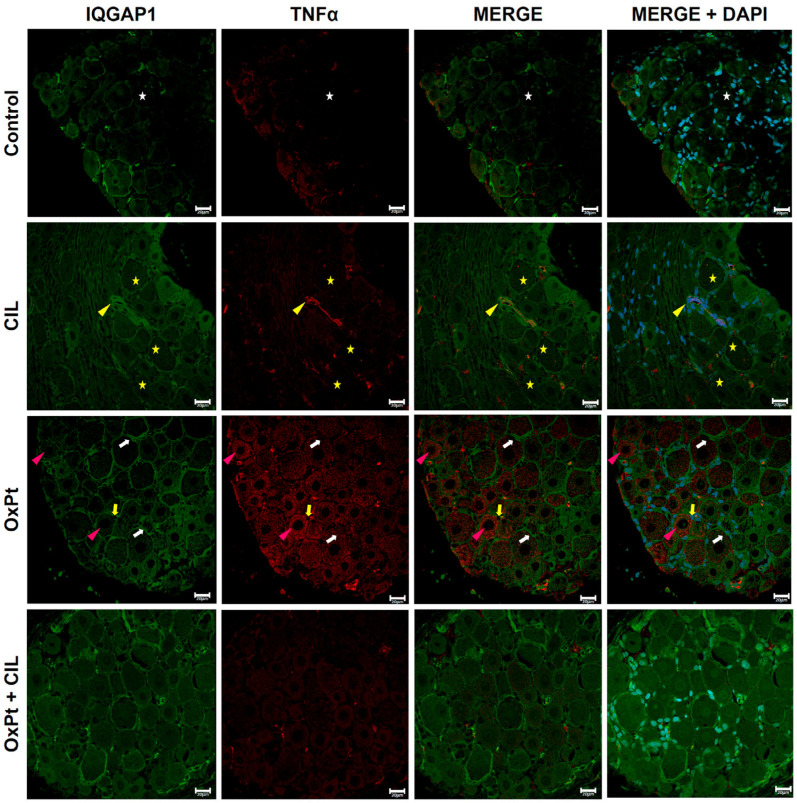
Immunostaining for IQGAP1 and TNFα. TNFα is absent or faint in neurons (white stars). IQGAP1 shows a heterogeneous distribution (as in Figure 3). Cilastatin (CIL): TNFα (yellow stars), in neurons and other cells (yellow arrowhead). OxPt: IQGAP1 in membrane and cytosol (white arrows). TNFα is strong in neuronal cytoplasm in a dot-like pattern (magenta arrowheads); the merge is heterogeneous. TNFα is expressed at medium-high levels in glia cells (yellow arrows). Scale bar 20 µm. OxPt + CIL: IQGAP1 and TNFα show distributions similar to controls. (*n*-Control = 4, 160 photographs, 7 DRGs; *n*-CIL = 4, 153 photographs, 7 DRGs; n-OxPt = 6, 297 photographs, 12 DRGs; *n*-Ox-Pt + CIL = 6, 217 photographs, 11 DRGs). OxPt increased TNFa expression in neurons and in glia 2–3-fold with respect to control and OxPt + CIL, *p* < 0.05. Appendix A includes higher-magnification images of representative regions from this figure.

**Figure 7 cells-14-01294-f007:**
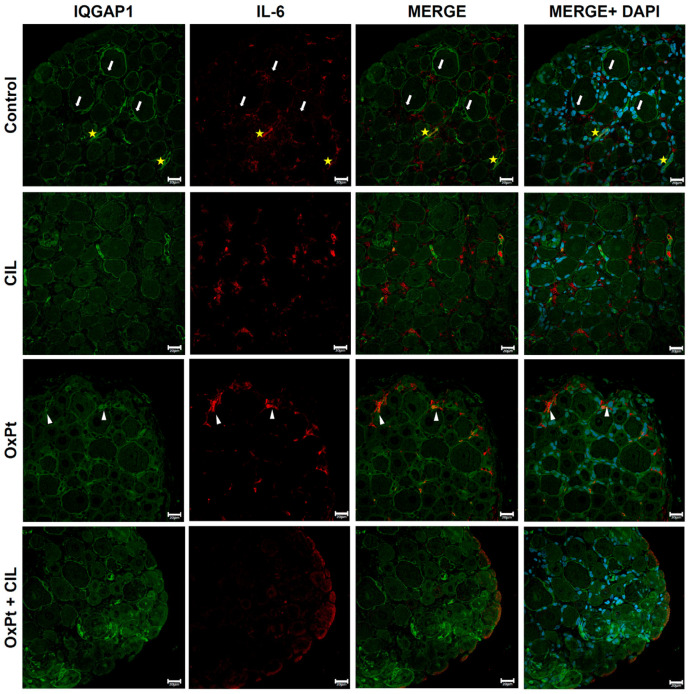
Immunostaining for IQGAP1 and IL-6. Control: IQGAP1 shows medium–high heterogeneous signal in neurons, glia, and capillaries (white arrows), outlining cell contours. IL-6 is low, with colocalization in capillary endothelial cells (yellow stars). Cilastatin (CIL): IQGAP1 and IL-6 signals remain comparable to controls. OxPt: IQGAP1 is medium and heterogeneous, lower than controls; increased in glia (white arrowheads). IL-6 colocalizes with IQGAP1 at high levels. OxPt + CIL: IQGAP1 remains high in membranes and perisomatic cytosol. IL-6 does not display a high-intensity signal. Scale bar 20 µm. (*n*-Control = 4, 183 photographs, 3 DRGs; *n*-CIL = 4, 200 photographs, 8 DRGs; n-OxPt = 6, 260 photographs, 10 DRGs; *n*-Ox-Pt + CIL = 6, 230 photographs, 8 DRGs). Enlarged views of key areas are shown in Appendix A.

**Figure 8 cells-14-01294-f008:**
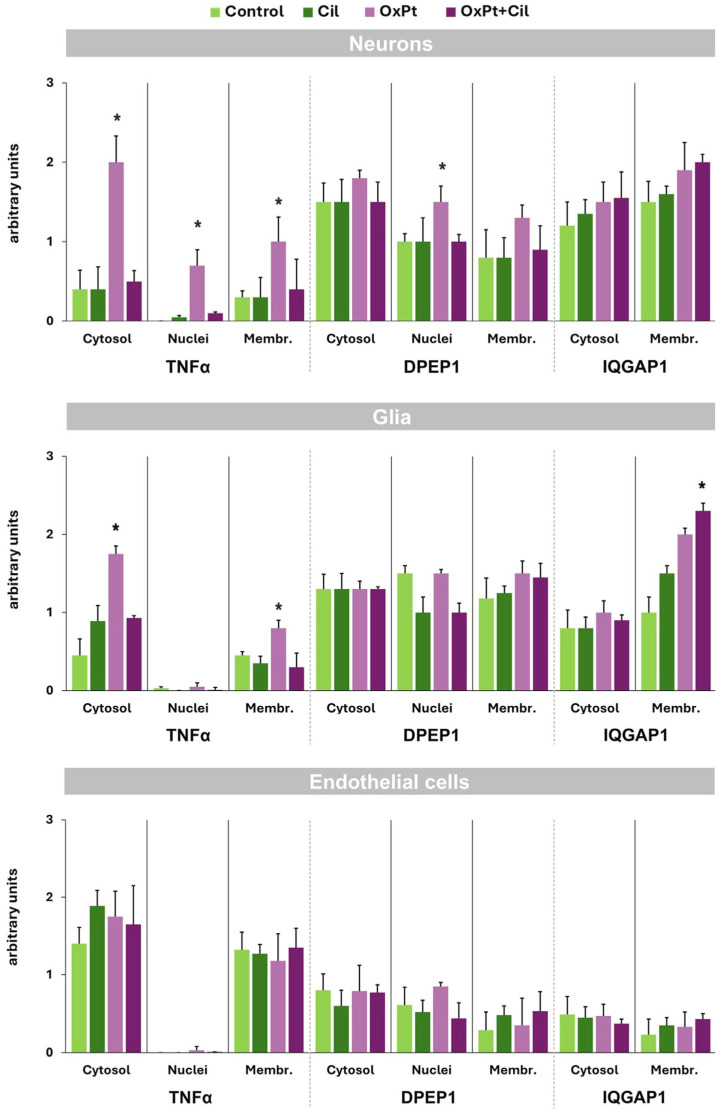
Intensity of specific fluorescence signal for TNF-α, DPEP1, and IQGAP1 in DRG neurons, glia, and endothelial cells in control, cilastatin (CIL)-, OxPt-, and OxPt + cilastatin-treated groups of rats. Neurons: a significant increase in TNF-α was observed in cytosol and nuclei of neurons in OxPt-treated animals, in contrast with basal levels of this inflammatory cell marker in the OxPt + CIL-treated group. The DPEP1 immunostaining signal was heterogeneous among the population of neurons. Heterogeneity in fluorescence intensities between cells reveals different response populations, which, however, when measured in all cells, hides possible statistical differences. There was a significant increase in DPEP1 signal in the nucleus of neurons of the OxPt-treated group versus basal levels in the rest of the groups, and a slight increase in IQGAP1 intensity signal in the cilastatin- and OxPt-treated groups compared with control. Glia: a significant increase in TNF-α was observed in the cytosol of satellite glia cells in OxPt-treated animals and the plasma membrane in the OxPt + cilastatin-treated group. The DPEP1 immunostaining signal was heterogeneous in the glial cell population. Endothelial cells: No statistical significance was found among the cells of the groups. Data are expressed as the mean ± SEM; * indicates statistical significance, *p* < 0.05.

**Table 1 cells-14-01294-t001:** Antibodies for immunohistochemistry used in this study.

Primary Antibodies					
Target		Host/Class	Dilution	Source	Cat.#
Anti-DPEP1		Rabbit polyclonal	1:100	Martín-Vasallo/Ávila [16]	DPEP1C
Anti-IQGAP1		Rabbit polyclonal	1:500	Millipore-Sigma Darmstadt, Germany	ABT186
Anti-IQGAP1		Mouse monoclonal	1:100	Santa Cruz Biotechnology Dallas, TX, USA	sc-376021
Anti-MAP2		Mouse monoclonal	1:500	Merck-Millipore	MAB378
Anti-GFAP		Mouse monoclonal	1:100	Santa Cruz Biotechnology Dallas, TX, USA	sc-33673
Anti-CD31		Mouse monoclonal	1:150	Santa Cruz Biotechnology Dallas, TX, USA	sc-376764
Anti-TNFα		Mouse monoclonal	1:150	Santa Cruz Biotechnology Dallas, TX, USA	sc-52B83
Anti-IL-6		Mouse monoclonal	1:200	Santa Cruz Biotechnology Dallas, TX, USA	sc-28343
Anti-Laminin		Mouse monoclonal	1:200	Santa Cruz Biotechnology Dallas, TX, USA	sc-365962
Anti-α-Actin		Mouse monoclonal	1:60,000	Sigma Aldrich/Merck Millipore Saint Louis, MO, USA/Darmstadt, Germany	A5441
**Secondary antibodies**					
**Target**	**Conjugation**	**Host/Class**	**Dilution**	**Source**	**Cat.#**
Rabbit IgG	FITC	Goat polyclonal	1:200	Sigma-Aldrich Saint Louis, MO, USA	F9887
Mouse IgG	DyLight^®^650	Goat polyclonal	1:100	Abcam Cambridge, UK	ab97018

## Data Availability

The data presented in this study are available on request from corresponding authors.

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
