# Peer review of "Cilastatin Modulates DPEP1- and IQGAP1-Associated Neuro-Glio-Vascular Inflammation in Oxaliplatin-Induced Peripheral Neurotoxicity"

_cells, 2025, doi:10.3390/cells14161294_

Round 1

Reviewer 1 Report

Comments and Suggestions for Authors

The manuscript titled, “Cilastatin modulates DPEP1- and IQGAP1-associated neuro-glia-vascular inflammation in oxiplatin-induced peripheral neurotoxicity” is an interesting study highlighting key players in inflammation of DRG and presents a potential modulator of OIPN. One of the strengths of the paper is demonstrating IQGAP1’s role in different cell types of the DRG milieu, particularly endothelial cells, glia, and neurons. Cilastatin seems to play a role via IQGAP1-mediated oxiplatin-induced inflammation. The authors had a comprehensive approach by looking at various cell types in this respect. However, the discussion section needs to be significantly expanded to bring home the main points the authors are stating and more importantly to connect the different components together. I would suggest beginning the first sentence of the discussion with a summary statement so the rest is easier to follow. At times portions in the manuscript, such as the introduction, are difficult to follow. It would greatly benefit if the authors include introductory statements before each paragraph for smoother flow and consistency. Please find below some of my comments:-

  1. In the introduction section, the segue from IQGAP1 to DPEP1 seems abrupt. Please include a connector statement or biological relevance for better flow.
  2. For Figure 2, please put a dark border around the western blots, otherwise it’s a bit difficult to differentiate between the blots. Additionally, I would highly recommend showing the statistics right next to the image, even in lack of significance.
  3. For figure 2, please include the housekeeping control such as tubulin or actin or GAPDH.
  4. For figures 3 & 4, it will really help to reduce the number of arrows and asterisks and include more higher magnification images to clearly demonstrate the differences. As it stands now, it’s quite difficult to gauge what the arrows are pointing at since it seems too crowded.
  5. For Figure 5, it’s hard to gauge what the magenta arrows and asterisks are pointing towards in the CD31 panel. The red fluorescence is faint in general.
  6. Please explain/amend 468-470. I’m not understanding the use of the term “the qualifying of puppeteer”. Is this meant to be a typo?

Reviewer 2 Report

Comments and Suggestions for Authors

The authors studied the expression of the enzyme dehydropeptidase-1 (DPEP1), the scaffold protein IQGAP1 and pro-inflammatory cytokines (TNF-α and IL-6) in neurons, glial and endothelial cells in dorsal root ganglia from rats with oxaliplatin-induced peripheral neuropathy. They also investigated the impact of cilastatin on the expression of the aforementioned molecules, assessing its potential therapeutic role in the treatment of chemotherapy-induced peripheral neuropathy. Thus, the study is important not only for theoretical science, but also because the obtained results may have significant clinical applications. The manuscript is well written and well organized. I have the following minor remarks:

In the Abstract I recommend using the term “neuro-glio-vascular unit” in place of “joint” e.g., “disruption of the function of neuro-glio-vascular unit”. (lines 28-29)‘

Introduction section – I proposed removing “particularly associated with platinum-based agents.”(lines 50-51), as in the first part of the sentence it is mentioned that it is an adverse effect of oxaliplatin which belongs to the group of platinum-based drugs. In addition, chemotherapy-induced peripheral neuropathy is not associated only with these agents. Many other anticancer drugs can induce severe, dose-limiting peripheral neuropathy.

The authors’results from previous studies on dehydropeptidase-1 (lines 115 116 and 119-120) and IQGAP1 (line 117-118) should be integrated into the respective paragraphs discussing their role in normal physiology and pathology. I recommend expanding the information on these previous findings.

Materials and methods – in the Results section, the authors describe the expression of DPEP1 and IQGAP1 in tissue homogenates from the colon, liver and kidney. However, the “Materials and methods” section does not mention the collection of these organs after the rats were euthanized.

Lines 165 – 168 – blood samples were collected for measuring biochemical parameters; however, such measurements were not performed in the experiment presented in the manuscript.

Line 175 – Is acetone applied to one hindpaw or both hindpaws? Later in the manuscript, it is refered to as “the tested paw” (lines 259-260). In the cited reference describing the method, acetone was applied to only one hindpaw.

Line 185 – Samples from which tissues were homogenized?

Results section

Line 255 – the terms “cold hyperalgesia” and “allodynia are not interchangable. Ïn the context of this manuscript, “allodynia” is the appropriate term to use.

The following sentence should be clarified for better understanding: “The counted number of biting, flicking or licking times of the tested paw increased by a factor of two to fifteen (9 to 15, day 5, Figure 1, panel B) in OxPt-treated rats relative to control and control+cilastatin-treated groups.”

The authors stated that “The number varied from 3-6 times/2 mins in controls to 8 to 18 in rats undergoing OxPt”, but based on fig. 1 the times for both controls are “zero”.

In the caption of fig. 1 should be clarified that the blue asterisk is in comparison to OxPt+cilastatin treated group and the red one is in comparison with both controls. Is there significant difference between controls and OxPt and OxPt+cilastatin treated groups from the second day. It is not mentioned in the text and it is not marked on the figure.

What is the rationale for examining the expression of DPEP1 and IQGAP1in the colon, kidney and liver? The sentence  ”However, when these tissues were studied by immuno histochemistry, remarkable differences…….” (lines 279-281) seems to be incomplete.

There are discrepancies between the manuscript text, figure captions, and the markers shown in the figures. For example, the caption of Figure 3 mentions white stars, white arrows, etc., but these are not present in the figure. I recommend that the authors carefully revise all figures in the manuscript and correct these discrepancies.

In addition, I advise the authors not to repeat the same information in both the figure captions and the main text, in order to avoid redundancy.

In the caption of fig, 8 (lines 452-453) is mentioned that “significant increase of TNF-α was observed in plasma membrane in OxPt+cilastatin treated group, but based on the fig. 8 and supplementary file it should be OxPt treated group. In addition, this is not indicated with asterisk on the figure.

The Discussion section should begin with а brief summary of the obtained results and their contribution to the researched scientific field.

Some abbreviations are defined several times in the manuscript, while others are not defined at all. Each abbreviation should be defined upon its first appearance in the text, and only the abbreviation should be used thereafter. I also suggest including  a list of all abbreviations at the end of the manuscript.

Reviewer 3 Report

Comments and Suggestions for Authors

In this manuscript, Martín-Ramírez et al. explore the neurological consequences of oxaliplatin-induced peripheral neurotoxicity, with particular emphasis on its effects in the dorsal root ganglia. The study examines the roles of IQGAP1 and DPEP1 proteins in mediating inflammatory responses within neurons, glial cells, and endothelial cells. Notably, the authors demonstrate the potential of cilastatin to attenuate oxaliplatin-induced neurotoxicity by stabilizing neuro-glio-vascular interactions and reducing inflammation. IQGAP1 and DPEP1 are identified as critical mediators in the pathogenesis of this condition, and cilastatin is proposed as a promising therapeutic agent. However, several major concerns regarding the experimental design, data validation, and interpretation of results need to be addressed. The following comments are provided for the authors’ consideration:

  1. In Section 3.1, the authors report that “Significant differences (p<0.05) were found from the second day on between the OxPt and OxPt+cilastatin treated groups, showing a protective effect of cilastatin from OxPt-elicited allodynia.” In the current study, the number of paw flicking, licking, and biting episodes in the control and OxPt+CIL groups was 0 and 2, respectively, while the OxPt group showed 10 episodes. However, in their previous study (PMID: 39278931), the control group exhibited 5 episodes and the OxPt group 10. Based on these values, the effect of cilastatin in the current study may simply fall within the variability range of the control group, rather than indicating a true therapeutic effect. This discrepancy raises concerns regarding data consistency and validation. It is possible that the apparent effect of cilastatin is not due to its pharmacological action, but rather a result of data fluctuation or statistical noise leading to a false-positive outcome. The authors should clarify this inconsistency and provide an explanation for the differing baseline responses observed between studies.
  2. In Section 3.2, the authors state that the quantification showed slight or no variation in intensity. However, there is no statistical evidence provided to support this claim. Qualitative descriptions alone are insufficient to substantiate the results. Furthermore, the presented Western blot lacks an internal loading control to confirm equal protein loading, which is essential for validating the findings. The authors are strongly encouraged to include an appropriate internal control and provide quantitative statistical analysis to support their hypothesis. This addition would significantly strengthen the credibility of the proposed interpretation.
  3. The authors demonstrated that IQGAP1-specific immunostaining signals showed higher intensity in satellite cells compared to the control group. However, the DRG contains up to seven different cell types. How do the authors confirm that the observed signal is specifically localized to satellite glial cells? The reviewer strongly recommends that the authors use a satellite glial cell marker, such as glutamine synthetase, to specifically identify these cells. By performing double-labeling with a satellite glial cell marker and IQGAP1, the authors could provide stronger evidence to support their hypothesis. A similar approach was well-executed in Section 3.4, where the authors effectively demonstrated IQGAP1 expression in glial/astrocytic cells by using GFAP as a specific marker.
  4. In Section 3.5, the authors stated that the brighter IQGAP1 signal around neurons corresponded primarily to satellite glial cells. However, using signal intensity alone is not a reliable method to distinguish cell types, as intensity can be affected by imaging parameters such as exposure time. The authors are encouraged to quantify the expression using objective methods, such as the percentage of positive cells or the number of positive cells per section or per ganglion, to provide statistically meaningful data. In the current study, the authors relied solely on qualitative descriptions, which are subjective and dependent on individual interpretation. This weakens the support for their hypothesis.
  5. In Section 3.6, the results showed that TNFα was detected in neurons. However, the reviewer questions why TNFα expression was not observed in satellite glial cells in the OxPt group, especially since oxaliplatin is known to increase TNFα levels in satellite glial cells. This discrepancy needs to be addressed and clarified. It is important because these results are used to support the authors’ claim regarding oxaliplatin-induced peripheral neurotoxicity.
  6. The authors provided a statistical chart to show differences between all groups. However, the reviewer questions how the arbitrary units were determined and what type of data were input into the statistical analysis to calculate these differences. Additionally, the authors used a semi-quantitative method, in which two independent observers evaluated the samples blindly and graded the staining intensity as absent (–), faint (+), moderate (++), or strong (+++). The reviewer asks whether this method has been validated as a reliable approach for statistical analysis, as the results may vary depending on the subjective judgment of the observers. This could lead to inconsistent or non-reproducible outcomes.
  7.  In the present study, the authors attempt to explain the differential distribution of DPEP1 and IQGAP1 across various cell types. However, they do not provide sufficient supporting evidence for this claim using immunohistochemistry alone. The reviewer recommends that the authors consider performing Western blot analysis with subcellular fractionation (cytosolic, nuclear, and membrane fractions) to better clarify the distinct localization of these proteins. Additionally, the reviewer suggests that the overall organization, experimental design, data validation, and interpretation of results require substantial revision to improve the quality and rigor of the manuscript.

The following minor issues should be addressed:

  1. In the Introduction (line 86), the authors state that decreased IQGAP1 expression impairs ………, but no references are provided to support this claim. Please include appropriate citations to substantiate this statement.
  2. Please spell out the abbreviation "GPI" in line 100 upon its first use.
  3. Please consider combining the first four paragraphs into a single paragraph to maintain coherence and present the related content as a unified discussion.
  4. The sentence in line 115 should be incorporated into the preceding paragraph to improve readability and logical flow.
  5. In lines 212–213, "300ppi" should be corrected to "300 dpi" (dots per inch), which is the appropriate term for image resolution in print and publication contexts.
  6. Several symbols mentioned in the figure legends are missing from the corresponding figures. Please ensure consistency between the figures and their legends.
  7. Additionally, there are numerous typographical errors throughout the manuscript. Please carefully revise the text for clarity and correctness.

Reviewer 4 Report

Comments and Suggestions for Authors

This manuscript by Rita Martín-Ramírez et al. presents a mechanistic investigation into the role of IQGAP1 and DPEP1 in oxaliplatin-induced peripheral neurotoxicity (OIPN), with a particular focus on dorsal root ganglion (DRG) inflammation and neuro-glio-vascular dysfunction. The authors elegantly combine behavioral and immunohistochemical analyses to characterize the cellular and subcellular distribution of IQGAP1 and DPEP1 in DRG neurons, glia, and endothelial cells, and explore the neuroprotective potential of cilastatin as a modulator of these pathways.

The study is well-structured and addresses an important clinical problem—chemotherapy-induced peripheral neuropathy—which currently lacks effective preventive or therapeutic options. The identification of IQGAP1 as a pleiotropic regulator of DRG inflammation and cytoskeletal integrity, along with its modulation by cilastatin, is particularly interesting. I believe that the results provided support the hypothesis that cilastatin can attenuate OxPt-induced neurotoxicity through stabilization of the neuro-glio-vascular unit.

  Overall, this manuscript tackles a timely and clinically relevant topic with methodological rigor and conceptual depth. With some revisions and clearer mechanistic insights, it has strong potential to make a meaningful contribution to the field of neuroinflammation and chemotherapy-induced neurotoxicity.

Mechanistic Link Between DPEP1 and IQGAP1

While the study highlights alterations in both DPEP1 and IQGAP1 expression in response to oxaliplatin and cilastatin treatment, the functional relationship between these two proteins remains incompletely defined. The current manuscript suggests potential crosstalk, but the mechanistic connection remains largely speculative. To substantiate this link, the inclusion of targeted mechanistic experiments—such as knockdown or overexpression of DPEP1 followed by assessment of IQGAP1 localization or activity using cultured cells—would be highly informative. These additional data would help determine whether DPEP1 directly regulates IQGAP1 function or operates through parallel pathways.

Discussion: Translational Relevance and Human Pathology Context

The study would benefit from a more explicit discussion of how the observed alterations in the rodent DRG microenvironment relate to known features of human OIPN pathology. For example, are IQGAP1 and DPEP1 expression patterns or polymorphisms known to differ in patients who develop severe OIPN? Addressing such questions or citing relevant clinical datasets would enhance the translational relevance and strengthen the argument for therapeutic targeting of these proteins in human neuropathy

Statistical Analysis:

It is recommended that the statistical analysis for Figure 1 be re-evaluated using two-way ANOVA, as this approach would better account for the interaction between multiple experimental variables. Additionally, for clarity and transparency, the specific statistical tests used should be explicitly stated in each figure legend. This will help readers more easily interpret the results and assess the robustness of the findings.

Paragraph Structure and Readability:
The current manuscript contains numerous single-sentence paragraphs, which detract from readability and logical flow. This issue is particularly evident in the Introduction and Discussion sections. Constructing a paragraph with only one sentence is rarely appropriate and should be avoided unless it serves a specific rhetorical purpose. The authors are strongly encouraged to apply the principles of paragraph writing by organizing related ideas and arguments into cohesive paragraphs, each centered around a clear main point. Improving paragraph structure in these key sections will significantly enhance the clarity, coherence, and overall quality of the manuscript.

Cellular Localization Clarification:

To improve interpretability of the immunofluorescence data, the authors should provide higher-magnification images of individual cell types. These images should clearly demonstrate how nuclear, cytosolic, and membrane compartments were identified and distinguished. This is particularly important for validating the subcellular localization of IQGAP1 and DPEP1 signals.

Sample Size in Immunostaining Experiments:

The number of biological replicates (n) used for each immunostaining experiment should be explicitly stated in the corresponding figure legends. Clear reporting of sample sizes is essential for evaluating the robustness and reproducibility of the immunohistochemical findings.

Round 2

Reviewer 1 Report

Comments and Suggestions for Authors

The authors have addressed the main concerns. Just a quick note on Figure 2: Although the authors have presented the statistics of the western blots right next to the image, it is sufficient to just include it in the figure legends rather than on the image itself. The best way to represent the statistics is to show a graph of the protein band intensities of the blot, measured by densitometry and normalized to a housekeeping protein. However, since the authors mentioned the housekeeping expression varies, it is understandable why the figures are presented this way. Overall, after the suggested changes, the manuscript reads well. 

Author Response

We appreciate once more the time and comments of the Reviewer. Figure 2 has been changed according reviewer´s advice.

Reviewer 2 Report

Comments and Suggestions for Authors

I would like to thank the authors for considering my recommendations. They have revised the manuscript accordingly and provided satisfactory responses to my questions. I have no further critical comments or suggestions.

Author Response

The authors thank you.

Reviewer 3 Report

Comments and Suggestions for Authors

The authors have addressed all the issues raised by the reviewer. Although the quality of the article still does not meet the review’s standards, it has improved compared to the previous version.

Author Response

The authors thank you.

Reviewer 4 Report

Comments and Suggestions for Authors

I have no concerns.

Author Response

The authors thank you.